# Antecedents and Consequences of Perceived Inclusion in Academia

**DOI:** 10.3390/ijerph19010431

**Published:** 2021-12-31

**Authors:** Siw Tone Innstrand, Karoline Grødal

**Affiliations:** Department of Psychology, Norwegian University of Science and Technology, 7491 Trondheim, Norway; karoline.grodal@ntnu.no

**Keywords:** diversity and gender in the workplace, leadership, mental health, perceived inclusion, work-life balance

## Abstract

A diversified workforce is a current trend in organizations today. The present paper illuminates the antecedents, consequences, and potential gender differences of a rather new concept salient to contemporary work life, namely, perceived inclusion. The hypothesized relationships were tested in a sample of academics and faculty staff at different higher education institutions in Norway (*n* = 12,170). Structural equation modeling analyses supported hypotheses that empowering leadership and social support from the leader (but not the fairness) are positively related to perceived inclusion. Further, perceived inclusion is positively related to organizational commitment, work engagement, and work–home facilitation and negatively related to work–home conflict. By utilizing multigroup analyses, we found support for the hypothesis that compared to women, men perceive their organization as more inclusive. However, in contrast to what was hypothesized, the proposed relationships in the model were stronger for men than women, suggesting that not only do men perceive their work environment as more inclusive, but their perception of inclusion is also more strongly related to beneficial outcomes for the organization. These results provide insight into the antecedents of and strategies for fostering an inclusive work environment, as a response to leveraging and integrating diversity in everyday work life.

## 1. Introduction

The changing nature of work life, characterized by globalization, immigration, worker migration, and the entry of more women and members of racial and ethnic minority groups, adds up to increased diversity, both in the European labor force and in the context of work in Europe [1]. Faculty demographics mirror these trends, with diversity in gender, ethnicity, race, family status, and age [2], also among students [3]. At the same time, social trends, including an aging population and more retirees than new entries to work, urge the need to retain and effectively utilize all employees. There is worldwide awareness that a sustainable society is dependent upon the inclusion of all individuals, regardless of gender, racial and ethnic background, religion, age, or physical foundation [4,5], and that inclusiveness is a prerequisite for creating a psychologically healthy workplace [6].

Examining the literature on workforce diversity, Saxena [7] concludes that diversity is a key to improved productivity if managed properly through multiple benefits, such as innovation, improved adoption in the global market, better problem-solving, creativity, and a varied collection of skills and experiences. A recent review suggests that a diverse workforce is also linked to quality of care, economic improvements, and better communication within staff [4]. However, simply being “diverse” is not enough. To utilize the strength and potential of a diverse workforce, the individual worker’s perception of inclusion is essential [8]. The distinction between diversity and inclusion is captured in a statement by Emily Hickey: “Diversity is inviting people to the party, whereas inclusion is asking them to dance, as they are able (p. 3)” [9]. Thus, while definitions of diversity focus on the demographic makeup of groups, inclusion emphasizes leveraging and integrating diversity into everyday work life [10]. Recent research supports the notion that diversity and inclusion are distinct constructs and even suggests that inclusion goes above and beyond diversity practices [11]. As inclusion has been linked to beneficial outcomes (i.e., job satisfaction, job performance, commitment, trust, self-reported helping behavior, and employee wellbeing), it has been suggested that inclusion may be a critical condition for organizations to realize the benefits of effective diversity practices [11,12,13,14]. Although there has been a great deal of research on workgroup diversity, there has been less research on inclusion [8]. While the former has focused on problems associated with diversity, inclusion offers a more positive approach by suggesting how the work environment can be created so diverse groups feel included.

By reviewing the inclusion and diversity literature, Shore et al. [8] have proposed a theoretical model of how inclusiveness provides an essential connection between different contextual antecedents (i.e., inclusive climate, inclusive leadership, and inclusive practice) and associated beneficial outcomes (i.e., job satisfaction, intention to stay, job performance, and organizational commitment). Building upon social exchange theory [15], they propose that fair treatment of employees associated with inclusion and an inclusive leadership style and practice would create a reciprocation to the organization manifested as organizational citizenship, commitment, and work performance.

The present study illuminates essential parts of this theoretical model by exploring how leadership affects the perception of inclusion and its corresponding impact on work–home conflict/facilitation, work engagement, and commitment. This is one out of a few studies exploring individual-level outcomes of inclusiveness. Moreover, gender differences are explored in both the level of each variable and their interrelationships. To the best of our knowledge, no empirical study has estimated the relationships in Shores et al.’s [8] model of inclusion simultaneously and tested for group invariance. The proposed model, integrating the determinants (i.e., empowering leadership, social support of the leader, and fairness of the leader) and consequences (i.e., commitment, work engagement, and work–home conflict/facilitation) of inclusion, is outlined in Figure 1.

### 1.1. Review of the Literature

#### 1.1.1. Perceived Inclusion

As the research on inclusion is still in the initial stages [16], there is limited agreement on the conceptual underpinning of the construct and how to define it [8,17]. Building upon Brewer’s optimal distinctiveness theory (ODT), Shore et al. [8] define inclusion as “the degree to which an employee perceives that he or she is an esteemed member of the workgroup through experiencing treatment that satisfies his or her needs for belongingness and uniqueness” (p. 1265). Thus, inclusion differs from related concepts such as social identification by suggesting that it is the group that includes the individual rather than the individual who connects to the group [18]. As the group is seen as the primary source of the perception of inclusion, inclusion should be assessed by measuring the climate for inclusion. Whereas inclusion refers to the individual’s sense of being part of the organizational system, a climate for inclusion refers to “employee perceptions of the organizational context that leads to the full acceptance of all employees and provides an environment in which the full spectrum of talents of individual employees are used” [19] (p. 5).

In the present study, perception of inclusion, or climate for inclusion, is operationalized and assessed by asking if there is room for unique groups in their unit (e.g., employees of different ethnic background or religion, older employees, or employees with various illness or disabilities) and if men and women are treated as equals. By means of multilevel analysis, Li et al. [20] found an inclusive climate to have a critical role in linking diversity management at the organizational level and individual affective commitment. Similarly, in their meta-analysis, Mor Barak et al. [19] found a climate for inclusion to be positively related to beneficial outcomes such as job satisfaction, commitment, decreased turnover intention, and better performance/productivity. Based on their review, they called for more studies on the potential factors and antecedents to enlighten our understanding of how to channel diversity into these beneficial organizational outcomes. We suggest that leaders play an important role in setting the tone for an inclusive climate by promoting equity, support, and empowerment in the workplace.

#### 1.1.2. Leadership

Leaders and leadership have been identified as key factors influencing the employee experience of inclusion [16]. Leaders play an important role in the employees’ perception of inclusion by promoting equity, voice, participation, and empowerment in the workplace and fostering an environment where everyone feels welcome and is treated equally [9]. However, despite a growing interest in inclusive leadership in general, there is limited research on how the dyad of the leader and employees affects feelings of inclusion and hence impacts beneficial outcomes for the organization in countries outside the US [16].

In a comprehensive study encompassing six different countries—Australia, China (Shanghai), Germany, India, Mexico, and the United States—Prime and Salib [21] wanted to explore what leadership behaviors can promote inclusion. Their findings suggest similarities in not only how employees characterize inclusion but also similarities in the leadership behaviors that foster it. More specifically, they found empowerment to be the strongest predictor for inclusion, followed by humility, courage, and accountability. This altruistic leadership behavior had a strong effect on employee innovation and team citizenship via inclusion of all countries and across genders.

Empowering leadership is defined as leaders’ actions to share influence with their employees, such as delegating authority to employees, promoting their self-directed and autonomous decision-making, coaching, sharing information, and asking for input [22]. Previous studies among academics have found empowering leadership to be positively related to work engagement through job autonomy, social community at work, and unreasonable tasks [23]. However, there were found no direct relationship between empowering leadership and work engagement. Similarly, Tuckey et al. [24] found empowering leadership and work engagement to be partially mediated by individual perceptions of working conditions. In the present study, we want to elaborate on these findings by exploring how an empowering leadership style is related to the perception of inclusion, as suggested by Shore et al. [8].

Moreover, as inclusion is rooted in fairness, equity, and social justice [25], fair and equitable treatment, particularly from leaders, seems to be significant [26]. In a review of the empirical literature in leadership and fairness, van Knippenberg et al. [27] conclude that fairness matters. Leaders who are more fair build better relationships with their followers, engender more positive attitudes, emotions, and more desirable and less undesirable behavior. Equity theory [28] builds upon the assumption that people value fair treatment, which causes them to be motivated to maintain fairness within the relationships of their co-workers. Therefore, we argue that employees’ sense of inclusiveness and equity may be derived from the social support and fairness of the leader.

Although there is a lack of empirical research on how fairness is related to inclusion in general, Shore et al. [8] argue that fairness should be integrated into their conceptual model as an antecedent due to its importance in establishing a climate for inclusion. Indeed, Kossek et al. [29] identified the leader’s action for fairness, talent leveraging, and workplace support as the most important factors for a climate of gender inclusion. The significance of trust and social support in an academic setting was highlighted by Vigoda-Gadot and Talmud [30], who found that the potentially negative aftermaths of perceived organizational politics can be controlled and reduced when trust and social support dominate the intra-organizational climate in one of Israel’s major research universities. Moreover, this mutual altruistic behavior was positively related to several beneficial job outcomes (i.e., job satisfaction, organizational commitment, stress, and burnout). Thus, in addition to an empowering leadership style, the present study aims to explore how support and fairness from the leader will provide a climate for inclusion perceived by the employees. More specifically, it is hypothesized that:

**Hypothesis** **1** **(H1).**
*Empowering leadership, social support from the leader, and fairness of the leader are positively related to perceived inclusion.*


#### 1.1.3. Organizational Outcomes

In line with Shore and colleagues’ proposal [8], and based on social exchange theory [15], one could expect a reciprocation of inclusive treatment in the organization manifested as beneficial organizational outcomes. Indeed, Cottrill et al. [10] found inclusive environments to promote employees’ work-related self-esteem and willingness to go above and beyond in their jobs. In the same line, Panicker et al. [17] found inclusive practices, inclusive climate, and inclusive leadership to be positively related to organizational citizenship behavior among academicians of a higher education institution in India. Closely related, a recent study by Auzoult and Mazilescu [31] found ethical climate as a social norm (i.e., as a set of rules perceived as applicable and expected by others) to be positively associated with the intention to rest, trust in a leader, and sociomoral judgment, and negatively to the propensity to discriminate.

The work of Findler et al. [13] suggests that employees who feel supported, included in decision-making, and treated fairly are more likely to report organizational commitment. Organizational commitment refers to the worker’s emotional attachment to, identification with, and involvement in the organization [32]. Examining the predictive value of job demands and resources on organizational commitment across different age groups in the higher education sector in Norway, Anthun and Innstrand [33] found empowering leadership to be positively related to the organizational commitment for all age groups. However, the relationship between perceived inclusion and organizational commitment is largely unknown, particular in an academic setting. One exception is Cho and Mor Barak’s [12] findings, which suggest that inclusion is significantly related to organizational commitment and job performance among Korean employees. Exploring the beneficial outcome of perceived inclusion on organizational commitment among academics in Norway, we hypothesize:

**Hypothesis** **2** **(H2).**
*Perceived inclusion is positively related to organizational commitment.*


In a similar way, Choi et al. [34] argued—and found support—that the reciprocation of support and feelings of inclusion could manifest in the organization through employees’ increased work engagement. Work engagement is mostly defined as “a positive, fulfilling, work-related state of mind (p.74)” [35], characterized and measured by three dimensions: vigor, dedication, and absorption [36]. These three dimensions refer to one’s energy, involvement, and ability to be deeply immersed in one’s work, respectively. To uncover whether these three dimensions have different causes and consequences, a differentiation between the three aspects are recommended [36]. Examining the relationship between diversity practices and work engagement, Downey et al. [11] found inclusion to be a significant moderator with a strong direct relationship to a trust climate. More specifically, their findings suggested that diversity practices were only positively related to a trusting climate, and hence work engagement, when employees perceived high levels of inclusion. Similarly, Nembhard and Edmondson [37] found psychological safety to mediate the relationship between leader inclusion and engagement in quality improvement work.

There is a lack of studies exploring the direct relationship between perceived inclusion and work engagement. Exploring the potential antecedents of engagement is particularly critical in knowledge-intensive workplaces such as higher education institutions where individuals are the primary bearers of knowledge, and the employees become the competitive parameter in the company [38]. Therefore, we propose the following hypothesis:

**Hypothesis** **3** **(H3).**
*Perceived inclusion is positively related to work engagement (vigor, dedication, and absorption).*


One corollary of a more diverse workforce and an aging population is the issue of work–home integration. A higher proportion of women participating in the workforce indicates that both men and women need to balance professional and family responsibilities; this is also highly evident in academia [2]. As the population ages, these family responsibilities might involve not only taking care of children but also the elderly. In addition, migration encompassing employees of different cultures and with different expectations and needs related to their family responsibilities demands leaders’ support and a perceived inclusive workplace. Indeed, work–home conflict is found to be particularly high among academics [39]. Work–home issues have been found to be the highest-ranked need for women in academics [40] and the strongest reason for women to consider leaving academic medicine [41]. There is now considerable evidence that work–home interactions can provide negative and positive experiences and feelings across the two domains conceptualized as work–home conflict and work–home facilitation or enrichment, respectively [42]. These two concepts are found to be independent and separate constructs with different antecedents and outcomes.

There is substantial indication that leader support plays a key role in the experience of work–home conflict [43]. However, the effect of perceived inclusion on work–home interaction is largely unexplored. One exception is a study by Brougham and Haar [44], which found perceived cultural inclusion to be both directly related to work–home conflict and enrichment, work–life balance, and indirectly related via perceived organizational support in a group of Mãori employees in New Zealand. In a similar way, we argue that perceived inclusion at work in our study would relate to work–home outcomes. More specifically, we propose:

**Hypothesis** **4** **(H4).**
*Perceived inclusion is negatively related to work–home conflict.*


**Hypothesis** **5** **(H5).**
*Perceived inclusion is positively related to work–home facilitation.*


#### 1.1.4. Gender Differences

Shore et al. [8] have argued that the salience of the need for belongingness and uniqueness, two central aspects for the perception of inclusion, is dependent on the contextual circumstances. For example, being a woman in a male-dominated working environment might activate the need for belonging if the woman feels disregarded in important decisions due to gender. Moreover, examining demographic similarity in the leader-subordinate dyad and family-supportive supervision, Foley et al. [45] found that leaders are more likely to empathize with similar subordinates (same gender and same race) and therefore provide more support to those subordinates on work–family issues. A meta-analysis of gender and science research conducted by the European Commission [46] indicates that women’s advancement in science is too slow, and the traditional view of science as gender-neutral is flawed. Hence, women’s salience of the need to belong and feel unique in a traditionally male-dominated workplace, combined with unequal opportunities and treatment, might induce female academics to perceive their work environment as less inclusive.

Moreover, a recent study by Salazar et al. [47] suggest higher vulnerability among women in academia. Studying the impact of the lockdown due to the COVID-19 they found higher scores in depression, anxiety, and stress among the female university workers as compared to the men. There is also some initial support for gender differences in the level of inclusion at the mean level. In a recent review on inclusive workplaces, Shore et al. [16] concluded that women, in general, appear to feel included less often than men. In a study among a group of electronic employees in the western United States, Mor Barak et al. [48] found that men and Caucasians perceived their organization as more inclusive than other groups. Similar findings have been reported in Israel [13], Korea [12], and Australia [20], whereas a study among academics in India provides a more fragmented picture, suggesting gender differences in the mean level of inclusive practices and leadership, but not in inclusive climate [17]. Therefore, we propose:

**Hypothesis** **6** **(H6).**
*Men perceive their organization as more inclusive as compared to women (mean level).*


However, gender differences in inclusion at the interrelation level—the strength of associations with other variables—is largely unexplored. Yet, building upon equity theory and the idea of social exchange and social reciprocity [15,28], one could assume that a stronger sense of inclusion may be derived from a sense that distribution of resources is fair to both relational partners and hence a stronger motivation to repay this equity with more commitment and engagement and better balance between work and family life. Thus, given that men might perceive their organization as more inclusive as compared to women, we hypothesized:

**Hypothesis** **7** **(H7).**
*The proposed relationships in the model (H1–H5) will be stronger for men than women (interrelation level).*


## 2. Materials and Methods

### 2.1. Design, Population, and Sample

Data in the present study is from KIWEST (Knowledge-Intensive Work Environment Survey Target) [49] used in the ARK intervention program [50]. ARK is the Norwegian acronym for work environment and climate study. The survey was sent by email to all employees in participating higher education institutions of the ARK study with regular payroll for a minimum 20% position (*n* = 18,599). Of those, 12,170 employees (65%) responded. About 38% had an academic position, 12% were doctoral research fellows, 45% were technical/administrative staff, and 5% had a position as a leader. The sample was quite equally dispersed across gender, with 54% women and 46% men. The age categories were distributed as follows: under 30 years, 9.8%; 30–39, 23.2%; 40–49, 27.2%; 50–59, 24.3%; and 60 years or older, 15.5%. Pearson’s chi-square test indicated significant differences between gender when it comes to age χ^2^(4) = 83.08, *p* = 0.000; condition of employment χ^2^(1) = 3.98, *p* < 0.05; and overtime χ^2^(3) = 281.43, *p* = 0.000. Demographics of the two subsamples were distributed as follows:

Women (*n* = 6527). Age was normally distributed: under 30 years (9%), 30–39 years (24%), 40–49 years (28%), 50–59 years (26%), and 60 years or older (13%). Most worked in a full position (83%) and had a permanent contract (74%). Overtime was reported frequently, as 53% reported working one to five extra hours, 19% reported six to ten extra hours, and 9% reported that they worked over ten hours beyond the agreed working hours per week.

Men (*n* = 5642). Age was normally distributed: under 30 years (11%), 30–39 years (22%), 40–49 years (26%), 50–59 years (23%), and 60 years or older (18%). Most worked in a full position (90%) and were permanently employed (76%). Overtime was reported frequently, as 43% reported working one to five extra hours, 25% reported six to ten extra hours, and 16% reported that they worked over ten hours beyond the agreed working hours per week.

### 2.2. Instruments and Variables

In this study, all response alternatives were given the same response alternatives, ranging from 1 (Strongly disagree) to 5 (Strongly agree). One exception is the engagement scale, where the response alternatives were 0 (Never), 1 (A few times a year or less), 2 (Once a month or less), 3 (A few times a month), 4 (Once a week), 5 (A few times a week), 6 (Every day).

#### 2.2.1. Antecedents

Empowering leadership from the General Nordic Questionnaire (QPS Nordic) [51] assesses employees’ perception of their management as empowering. This scale comprises three items, such as “My immediate superior encourages me to participate in important decisions”.

Social support leader from the Copenhagen Psychosocial Questionnaire (COPSOQ II) [39] has three items, such as “My immediate superior listens to me when I have problems at work”.

Fairness of the leader is a modified version of QPS-Nordic [51]. Leader’s fairness was measured by three items, such as “My immediate superior distributes work assignments fairly”.

#### 2.2.2. Perceived Inclusion

A four-item scale from COPSOQ II [39] measured inclusion. The scale measures the perception of gender equality and inclusion for older employees, employees of different ethnic background or religion, and employees with various illnesses or disabilities. Sample items are “Men and women are treated as equals in my unit” and “In my unit, there is room for older employees”.

#### 2.2.3. Outcomes

Commitment is measured by three items, such as “I am happy to tell others about my workplace”, and captures the respondents’ positive ties to their workplace. The scale is a revised version of the four-item scale from COPSOQ II [39] and assesses affective commitment. One of the original items was omitted due to a low factor loading (=0.35) [49].

Work engagement was assessed by three dimensions: vigor (i.e., “At my work, I feel bursting with energy”), dedication (i.e., “I am proud of the work that I do”), and absorption (i.e., “I get carried away when I’m working”). Each dimension comprises three items each and is from the shortened version of the Utrecht Work Engagement Scale (UWES). Although a one-factor model of engagement can be justified by high correlations between the three dimensions, confirmatory factor analyses have shown that the hypothesized three-factor structure of the UWES is superior to the one-factor model [36].

Work–home conflict (WHC) and work–home facilitation (WHF) were measured by four items each. Sample items are, “Stress at work makes me irritable at home” (WHC) and “Having a good day at work makes me a better companion when I get home” (WHF). This scale is a slightly modified version [42] of the scale from Wayne et al. [52].

### 2.3. Analyses

Stata version 14.2 (Stata Statistical Software: Release 14. College Station, TX, USA: StataCorp LP) was used to analyze the data with structural equation modeling (SEM). All analyses were performed with maximum likelihood estimation with missing values (method option MLMV in Stata), which means that information from observations with missing values was included in the analyses. As the χ^2^ is sensitive to sample size (large samples increases the probability of model rejection), the following goodness-of-fit indices and cut-offs were used in all analyses: RMSEA (root mean squared error of approximation) <0.08, CFI (comparative fit index) >0.90, and TLI (Tucker–Lewis index) >0.90 [53]. Maximum likelihood estimation assumes normally distributed data. In this study, all values of univariate skewness and kurtosis were below 2.0 and 7.0 (Table 1), respectively, indicating no problems with non-normality [54]. However, Mardia’s [55] tests of multivariate skewness (7.29, *p* < 0.001) and kurtosis (152.08, *p* < 0.001) were significant. Therefore, the Satorra-Bentler (S-B) scaled χ^2^ test [56] with corresponding RMSEA, CFI, and TLI values, which are robust to non-normal data, were reported when possible (Stata did not allow for this in the multigroup analyses). Invariance between gender was evaluated using Chen’s [57] suggested cut-off values for change in the following fit indices: ∆RMSEA ≥ 0.015 and ∆CFI ≥ −0.010.

## 3. Results

### 3.1. Confirmatory Factor Analyses

As a first step, confirmatory factor analyses (CFA) were performed on the whole sample to investigate the suitability of the measurement model. The modification indices (MI) suggested that the model fit could be improved substantially by allowing some of the error terms between two items within work engagement and work–home conflict to correlate. Inspection of the wording of these items justified their similarity, and their error terms were allowed to correlate. More specifically, in engagement, the MI suggested correlated error terms for two of the items representing the vigor dimension (“At my work, I feel bursting with energy” and “At my job, I feel strong and vigorous”) and two items related to absorption (“I am immersed in my work” and “I get carried away when I’m working”). In work–home conflict, two items related to energy and effort were allowed to correlate (“My job reduces the effort I can give to activities at home” and “My job makes me feel too tired to do the things that need attention at home”) and two items related to mental issues/feelings (“Stress at work makes me irritable at home” and “Job worries or problems distract me when I am at home”).

The factor loadings were satisfactory (>0.50), with loadings from β = 0.55 to β = 0.91, except item 3 (“Having a good day at work makes me a better companion when I get home”), in work–home facilitation (β = 0.21), which was removed from further analyses. The goodness-of-fit estimates for the final measurement model all suggest a strong model fit (χ^2^ (df) = 7787.818 (415), *p* < 0.001, RMSEA = 0.038, CFI = 0.97, and TLI = 0.96; S-B χ^2^ (df) = 5697.560 (415), *p* < 0.001, S-B RMSEA = 0.034, S-B CFI = 0.97, and S-B TLI = 0.96).

Table 1 displays the mean, standard deviation (SD), skewness, kurtosis, correlations, composite reliability (CR), and Cronbach’s alpha (α) for the factors in the final measurement model. All factors were reliable with composite reliabilities and α coefficients above the cut-off of 0.7 [58], except inclusion (α = 0.67; CR = 0.67), yet the coefficient is a bit higher than found in the COPSOQ II (α = 0.63) [39]. Since this measure assesses different aspects of inclusion (i.e., room for elderly, men and women are treated as equals), the somewhat lower internal reliabilities were rather expected. As 0.67 is close to the suggested threshold of 0.70 and deleting any of the four items would not increase the reliability, the scale was kept unchanged.

### 3.2. Full Structural Equation Model

Hypotheses 1–5 were tested by running SEM with the hypothesized relationships as visualized in Figure 1. The SEM analysis provided an acceptable fit to the data (χ^2^ (df) = 20,510.470 (448), *p* < 0.001, RMSEA = 0.061, CFI = 0.914, TLI = 0.914; SB-χ^2^ (df) = 14,977.006 (448), *p* < 0.001, S-B RMSEA = 0.054, S-B CFI = 0.91, S-B TLI = 0.91). All hypothesized relationships, except fairness of the leader, were significant and in the expected direction. The strongest relationships were between the perceived inclusion and engagement dimensions (vigor, dedication, and absorption).

### 3.3. Multigroup Analyses

#### 3.3.1. Differences in the Mean of the Latent Variables

Before differences in the mean of the latent variables could be tested, the measurement model was tested separately among women and men. The results suggested a good model fit in both groups (Table 2). Invariance of the measurement model was further tested by running multigroup CFAs (MG-CFA) with three levels of constraints: (1) an unconstrained model; (2) equal factor loadings; and (3) equal factor loadings and intercepts. Although the chi-square increase was significant for each nested model, the preceding models were no worse, as the change in CFI did not exceed −0.01, and the change in the RMSEA was less than 0.015. [57].

As measurement invariance (loadings and intercepts) was established between gender, the differences in latent means could be calculated [59]. Hypothesis 6 was then tested by conducting MG-CFA on the measurement model, with factor loadings and intercepts constrained as equal. The reference group was set as women (latent mean fixed to 0). A significant mean among men, therefore, reflects that the genders score significantly different from each other on the specific variable. Table 3 provides the comparisons of the latent means on the study variables for women and men. Supporting Hypothesis 6, the average man’s score on perceived inclusion is 0.22 standard deviations above the score of the average woman. Although not hypothesized, Table 3 suggests that men report a significantly higher level of empowering leadership, social support from the leader, and fairness of the leader as compared to women. However, they report significantly less vigor and work–home interaction (conflict and facilitation). There are no gender differences in the level of commitment, dedication, or absorption.

#### 3.3.2. Differences in the Strength of the Associations

To test whether the structure between the latent variables is invariant or not across gender (Hypothesis 7), a multigroup SEM (MG-SEM) across gender was run on the same structural model as for Hypotheses 1–5 (Figure 1). The model fit of a constrained MG-SEM model with unstandardized parameters constrained to be equal for men and women (universal model) was compared to an unconstrained MG-SEM model where the parameters were allowed to vary between genders (group-sensitive model). Unstandardized values were inspected in the group comparisons [53], and the Wald test was used to evaluate the significance of group differences in the structural paths.

The group-sensitive model had a significantly better fit than the universal model according to differences in the chi-square value (Δχ^2^ (df) = 22.55 (9), *p* < 0.01). The other fit indices were left unchanged. Thus, gender differences in the strength of the relationship in the hypothesized structural model can be assumed as suggested by Hypothesis 7. More specifically, as indicated by Wald tests (significant differences marked in bold in the unconstrained model, Table 4), men reported inclusion to be stronger related to commitment (χ^2^ (df) = 8.333 (1); *p* < 0.01) and work engagement (vigor: χ^2^ (df) = 9.369 (1); *p* < 0.01; dedication: χ^2^ (df) = 9.250 (1); *p* < 0.01; absorption: χ^2^ (df) = 6.067 (1); *p* < 0.01) as compared to women. In general, the R^2^ in the unconstrained model suggests that inclusive leadership (empowering, supporting, and fair) accounted for about 17 and 17 percent each of the variance in inclusion for females and males, respectively. Equivalently, inclusion explains (female/male): 55/55 percent in commitment; 89/85 percent in vigor; 94/94 percent in dedication; 78/78 percent in absorption; 39/34 percent in work–home facilitation; and 17/15 percent in work–home conflict.

## 4. Discussion

A changing work life characterized by a more diverse workforce urges the need for knowledge on how to create inclusive work environments. The present study responds to this need by providing empirical research on how the behavior of leaders affects the feeling of inclusion, which has beneficial outcomes for both the individual and the organization.

More specifically, Hypothesis 1, which suggests that empowering leadership, social support from the leader, and fairness of the leader are positively related to inclusion, was partly supported. The SEM analyses suggested significant paths from empowering leadership and social support from the leader, but not from the fairness of the leader. The lack of a significant relationship between the fairness of the leader and perception of inclusion is in line with a recent study by Chung et al. [60], suggesting overall justice not to be significantly related to inclusion. The authors suggest that this surprising result can be due to a positive relationship between the antecedent variables. This could also be the case in the present study, as we used three highly related aspects of leadership behavior (Pearson r ranging from 0.75 to 0.85, see Table 1). Knippenberg et al. [27] propose that, in a sense, leader fairness may substitute for other aspects of leadership or vice versa, as in this case, and other aspects of leadership may substitute for leader fairness stealing all the variance of fairness. It should also be noted that leadership in academia might differ from other organizations, as it has employees with tenure and academic freedom to investigate self-chosen subjects. Thus, distributive fairness might have less importance in a highly autonomous work environment such as a university.

In line with the idea of social exchange and social reciprocity [15], inclusion was positively related to organizational commitment, work engagement (vigor, dedication, and absorption), and work–home facilitation, and negatively related to work–home conflict, supporting Hypotheses 2–5. This applied for the whole sample and across gender. In particular, inclusion is strongly related to work engagement. Thus, arranging for an inclusive work environment might have beneficial outcomes for both the individual and the organization. For example, a recent study conducted among Norwegian academic employees suggested that work engagement is related to productivity as measured by an increase in publication points on an aggregated level [61]. Given the clear importance of perceived inclusion and work engagement and commitment, an important question for future research pertains to which organizational variables may boost employees’ perception of inclusion, as empowering and supportive leadership only contributes to a small degree. This was a surprising result since previous studies have identified leaders and leadership as a key factor influencing the employee experience of inclusion [16]. Potential explanations for these weaker relationships found in the present study could relate to different measures of inclusion or because academic management might differ from other organizations.

It should be noted that perception of inclusiveness was treated as a reflective measure and modeled as a latent variable in the present study. However, as the scale items point to inclusiveness in relation to different characteristics (religion, age, etc.), which are not necessarily supposed to measure the same latent factor (e.g., a workplace might be not inclusive when it comes to age, but inclusive in relation to religious preferences), it could also be considered as a formative measure. Diamantopoulos, Riefler, and Roth [62] suggest that an incorrect reflective specification might provide structural paths that are either overestimated or underestimated if the misspecified variable is an exogenous or endogenous variable, respectively. However, due to several shortcomings and restrictions related to the use of formative measures in structural equation modeling, there is an ongoing debate of its usefulness [62]. Thus, we urge future researchers to develop and test sound measures of perceived inclusion that are valid, reflective, and at the same time capture all aspects of the concept. More research on the important determinates of perceived inclusion in an academic setting is also necessary.

In line with previous findings from other countries and settings [12,13,20], men at Norwegian universities perceive their organization as more inclusive as compared to their female colleagues, supporting Hypothesis 6. Although Norway has the highest proportion of women in higher education in Europe (45%), it is also among the European countries where the gender pay gap is higher in male-dominated occupations, possibly suggesting a situation where the organizational culture shows resistance toward integrating women [46]. Thus, unawareness of the barriers associated with being a member of a minority group might create a perception of more inclusion and fairness among men, as suggested by Mor Barak et al. [48]. As an unawareness of these barriers might prevent effective strategies for promoting an inclusive climate, the first challenge is for men to recognize that sexism or inequalities might exists. Thus, coaching females or disadvantaged groups to be more confident may only be part of the solution. Instead, Sawyer and Valerio [63] argue that to reduce systematic bias organizations should engage male leaders in gender-inclusive leadership by combining best practices in mentoring with an ally mentality, called “male champion”. Similarly, Li et al. [20] found identity-conscious programs (programs that target specific identity groups) to be effective in promoting an inclusive climate. Overall, Francis and Michielsens’ findings [64] suggest that inclusive companies have more female employees and leaders and features significantly higher mentoring and organizational training levels than exclusive companies.

The gender-sensitive model was superior to a generic model, suggesting possible gender differences in the strength of the relationship in the hypothesized model. In line with Hypothesis 7, men reported a stronger relationship between inclusion and commitment and work engagement (vigor, dedication, and absorption) than their female counterparts. Thus, men perceive their work environment as more inclusive; this perception of inclusion is also strongly related to beneficial outcomes for the organization among men. This relates to the findings among academics in India, where the relationship between inclusive climate and organizational citizenship behavior (OCB) appeared to be stronger for men (b = 0.32) than for women (b = 0.04) [17]. However, the regression in the Indian study was conducted in two separate analyses, and the significance of these potential gender differences could not be tested. The present study adds to this finding by suggesting significant gender differences in the strength of this relation.

There were no gender differences in the relationship between empowering leadership, social support from the leader, fairness of the leader, and inclusion, or between inclusion and work–home interaction (conflict and facilitation).

All in all, the hypothesized relationships were significant and in intended direction for both men and women, with one exception being fairness of the leader. This implies that an empowering and supportive leader would be beneficial for both men’s and women’s perception of an inclusive work environment, which is positively related to their commitment, work environment, and work–home interaction.

### 4.1. Limitations and Future Directions

The study findings are strengthened by the use of a large sample of university staff (*n* = 12,170). However, the study findings should be interpreted with some limitations in mind.

The first concern relates to using self-reported data, which implies a certain risk that the findings are based on common-method variance [65]. Next, our research was conducted on academics and staff at higher education institutions in Norway. Although the present study aligns with related findings in other countries, Norway, Finland, and Sweden are known as “global gender equality leaders”, committing efforts to embed gender equality into science policy and society at large since the late 1970s [46]. Thus, some gender differences revealed in the present study might be even stronger in countries where gender equality is less evolved. Moreover, the strength of the relationships found in the model might fluctuate in different contexts and organizations where diversity issues are less valued or salient. Nevertheless, the present study adds to the knowledge on inclusion outside the US and calls for more studies from different nations with varied legislative, social, and historical contexts [16].

In the present study, perceived inclusion was measured by adding four sub-components into one latent variable. Although the aim of the present study was to investigate climate for inclusion, dividing each sub-component could have provided different results and a wider spectrum of inclusion aspects. In addition, inclusion was measured with only surface-level diversity characteristics (gender, age, disability, and ethnicity). Future studies should explore a climate for inclusion based on deep-level diversity characteristics, those less immediately visible to others (e.g., education and job tenure). This might be particularly relevant in the educational sector, with its high level of professional hierarchy. Unfortunately, with the present dataset, we were not able to provide diversity characteristics such as ethnicity and disability. This could have further illuminated the context of investigation. Moreover, three aspects of leadership and leadership behavior were used as determinants for inclusion: empowering leadership, social support, and fairness of the leader. Although these leadership measures differ conceptually, the intercorrelation among them was rather high (r < 0.85) yet below the threshold of 0.90, which may indicate multicollinearity. Still, some explained variance might have been lost in this leadership triangle.

A maximum likelihood estimator was applied in this study, despite significant multivariate skewness and kurtosis. Some researchers have argued that this estimator is fairly robust to non-normally distributed data [66]; however, this might have caused a negative bias in parameter estimates and standard errors [67]. Other estimation methods such as robust maximum likelihood and asymptotic distribution-free estimation are recommended in cases of non-normality. However, such estimators in Stata were not applicable for the multigroup analyses or to obtain the needed goodness-of-fit indices.

Finally, the cross-sectional nature of this study limits our ability to make causal inferences from the data. Although the hypotheses tested in this study were based on a theoretical inclusion model [8] and existing literature, reversed causality cannot be ruled out on the basis of our results. Although the present study suggest that more inclusive environment fosters organizational commitment, an alternative explanation could be that an organization that has more committed employees is one that is also more inclusive. Future studies should use a longitudinal or experimental design to assure causal inference of the hypothesized relationships.

### 4.2. Implications

The present study suggests that a climate for inclusion in the workplace is a prerequisite for a sustainable organization, as it is positively linked to beneficial outcomes for organizations. This applies especially to the university setting, where the health and motivation of the workers are critical to delivering a high-quality service [50]. This, together with the trend of increased diversity in the faculty demographics [2], high level of work–home conflicts, and need for a balance found among academics [39], substantiate the importance of the findings in this study. Moreover, the present study contributes to occupational psychology literature by exploring a rather new and timely topic: perceived inclusion.

Building upon the theoretical assumptions made by Shore et al. [8], the present study tests and gives empirical support to most of the relationships they proposed. Our study also extends their model by suggesting that a perception of inclusion might have beneficial effects beyond the work environment, as it was positively related to work–home facilitation and negatively related to work–home conflict. In general, this is one out of few studies exploring individual-level outcomes of inclusion [60].

Practically, our results open new pathways for organizations to promote engagement, commitment, and work–home balance by facilitating an inclusive work environment. For example, as suggested by the present study, organizations should not underestimate the role of leaders in shaping a climate of inclusion. As gatekeepers of important initiatives for flexible arrangements for marginalized groups and an ideal for how to include all in the work environment, superiors have an important role in optimizing inclusion at work. Supported by the findings in our study, an empowering and supportive leadership style seems to be particularly essential in this matter. As suggested by Tuckey et al. [24], there are two broad options to achieve this end, either focusing on these qualities in the leadership recruitment process or during leadership training. For example, the ADVANCE leadership program at the University of Washington aims to provide department chairs the skills, community, and information needed to be agents of change within the university. Recently, this program has successfully expanded from campus-based workshop programs to national workshops (LEAD) to a web-based toolkit (LiY!), equipping department chairs to be advocates of gender equity, diversity, and inclusion [68].

Based on a recent review, Shore et al. [16] suggests two processes to foster inclusive workplaces by management, one where the manager has a prevention orientation (compliance with practices and policies) and one where the manager has a promotion orientation. In the latter, practices are related to promoting psychological safety, involvement in the workgroup, feeling respected and valued, influence on decision-making, authenticity, and recognizing, honoring, and advancing diversity. In addition, Travis et al. [25] have offered an overview of corporative practice examples that have advanced inclusion. As suggested by the present study, the successful outcome of such practices in boosting the perception of inclusion should be a more dedicated, energetic, and committed workforce. We hope the present study inspires more research and encourages organizations and leaders to support diversity and endeavor to foster an inclusive work environment for all.

## Figures and Tables

**Figure 1 ijerph-19-00431-f001:**
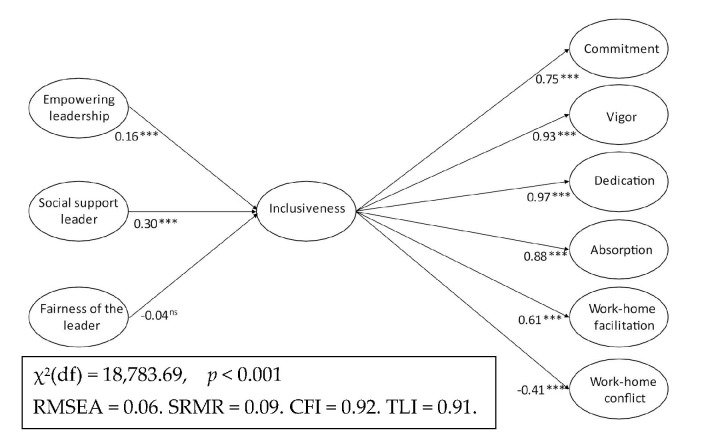
Structural model with standardized parameter estimates. Note: ***: *p* < 0.001; ^ns^ = not significant.

**Table 1 ijerph-19-00431-t001:** Pearson’s correlations and scale reliabilities between the factors in the measurement model.

Variable	*M*	*SD*	Skewness	Kurtosis	1	2	3	4	5	6	7	8	9	10
Inclusion	4.10	0.65	−0.73	3.99	−									
Empowering leadership	3.74	0.94	−0.74	3.32	0.44	−								
Social support from leader	3.73	0.92	−0.75	3.42	0.43	0.83	−							
Fairness of the leader	3.74	0.89	−0.65	3.39	0.49	0.75	0.77	−						
Commitment	3.93	0.75	−0.84	4.04	0.43	0.52	0.50	0.49	−					
Vigor	4.71	1.07	−1.38	5.12	0.22	0.29	0.30	0.25	0.52	−				
Dedication	4.80	1.17	−1.32	4.69	0.24	0.34	0.33	0.29	0.61	0.76	−			
Absorption	4.30	1.26	−0.95	3.60	0.26	0.24	0.22	0.20	0.46	0.60	0.73	−		
Work–home facilitation	3.20	0.62	−0.14	3.53	0.21	0.32	0.30	0.26	0.49	0.36	0.43	0.34	−	
Work–home conflict	2.97	0.87	−0.03	2.57	−0.26	−0.25	−0.29	−0.30	−0.33	−0.34	−0.27	−0.10	−0.20	−
CR					0.67	0.90	0.86	0.84	0.82	0.77	0.91	0.71	0.70	0.75
A					0.67	0.89	0.85	0.84	0.80	0.87	0.90	0.85	0.70	0.82

CR: composite reliability; α: Cronbach´s alpha. All significant at *p* < 0.001.

**Table 2 ijerph-19-00431-t002:** Tests of factorial invariance of the measurement model across genders (*n* = 12,168).

	χ^2^ (df)	RMSEA	CFI	TLI	∆RMSEA	∆CFI	∆χ^2^ (df)
Single-group solutions							
Women (*n* = 6526)	4428.929 (415) *** (3239.442 (415) ***)	0.039 (0.034)	0.968 (0.969)	0.962 (0.963)			
Men (*n* = 5642)	3922.079 (415) *** (2920.792 (415) ***)	0.039 (0.034)	0.968 (0.968)	0.961 (0.962)			
Measurement invariance							
Equal forms (unconstrained)	8351.007 (830) ***	0.039	0.968	0.962			
Equal factor loadings	8424.299 (852) ***	0.038	0.968	0.962	0.000	0.000	73.291 (22) ***
Equal factor loadings and intercepts	9243.950 (884) ***	0.039	0.964	0.960	−0.004	−0.002	819.651 (32) ***

*** *p* < 0.001. Results from analysis with the Satorra–Bentler estimator in brackets.

**Table 3 ijerph-19-00431-t003:** Comparison of the means of the study variables for women and men.

Variable	Women		Men		Pooled SD	Effect Size
	Latent Mean	Variance/SD	Latent Mean	Variance/SD		
Inclusion	0	0.37/0.61	0.13 ***	0.31/0.56	0.59	0.22
Empowering leadership	0	0.98/0.99	0.05 **	0.96/0.98	0.99	0.05
Social support from leader	0	0.74/0.86	0.08 ***	0.68/0.82	0.84	0.10
Fairness of the leader	0	0.75/0.87	0.16 ***	0.66/0.81	0.84	0.19
Commitment	0	0.57/0.75	0.01	0.57/0.75	0.75	0.01
Vigor	0	0.67/0.82	−0.05 **	0.71/0.84	0.83	−0.06
Dedication	0	1.23/1.11	−0.02	1.26/1.12	1.12	−0.02
Absorption	0	1.23/1.11	−0.01	1.18/1.09	1.10	−0.01
Work–home facilitation	0	0.42/0.65	−0.05 ***	0.41/0.64	0.65	−0.08
Work–home conflict	0	0.43/0.66	−0.11 ***	0.41/0.64	0.65	−0.17

Reference group: women. Measurement model with loadings and intercepts constrained to be equal for women and men. Effect size = Latent mean/pooled standard deviation. ** *p* < 0.01. *** *p* < 0.001.

**Table 4 ijerph-19-00431-t004:** Path coefficients of the multigroup structural equation (MG-SEM) model, by gender.

	Universal Model (Constrained Solution)				Group-Sensitive Model (Unconstrained Solution)			
	Female (*n* = 6526)		Male (*n* = 5642)		Female (*n* = 6526)		Male (*n* = 5642)	
	*B*	β	*B*	β	*B*	β	*B*	β
Contextual antecedents:								
Empowering leadership→Inclusion	0.04 ***	0.16 ***	0.04 ***	0.16 ***	0.03 **	0.13 **	0.05 ***	0.20 ***
Social support from leader →Inclusion	0.09 ***	0.31 ***	0.09 ***	0.29 ***	0.09 ***	0.29 ***	0.09 ***	0.32 ***
Fairness of the leader→ Inclusion	−0.13	−0.05	−0.13	−0.04	0.00	0.00	−0.03 **	−0.12 **
Outcomes:								
Inclusion→Commitment	2.35 ***	0.74 ***	2.35 ***	0.74 ***	2.17 ***	0.74 ***	2.52 ***	0.74 **
Inclusion→ Vigor	3.21 ***	0.94 ***	3.21 ***	0.92 ***	2.95 ***	0.94 ***	3.45 ***	0.92 ***
Inclusion→ Dedication	4.51 ***	0.97 ***	4.51 ***	0.97 ***	4.14 ***	0.97 ***	4.84 ***	0.97 ***
Inclusion→ Absorption	4.11 ***	0.88 ***	4.11 ***	0.88 ***	3.82 ***	0.88 ***	4.34 ***	0.88 ***
Inclusion→Work–home facilitation	1.61 ***	0.61 ***	1.61 ***	0.60 ***	1.52 ***	0.62 ***	1.65 ***	0.59 ***
Inclusion→Work–home conflict	−1.12 ***	−0.40 ***	−1.12 ***	−0.41 ***	−1.08 ***	−0.42 ***	−1.14 ***	−0.39 ***
χ^2^ (df)	21,921.77 (956) ***				21,899.22 (947) ***			
∆χ^2^ constrained solution vs. unconstrained solution	22.55 (9) **							
RMSEA	0.06				0.06			
CFI	0.91				0.91			
TLI	0.91				0.91			
SRMR								

All structural parameters were fixed to be equal in the constrained solution. Bold text indicates significant group differences as indicated by Wald tests for group invariance (*p* < 0.05)—unconstrained model. Group comparisons made on the unstandardized coefficients (*B*). R^2^ (female/male): inclusive leadership (empowering, supporting, and fair) 17/17 percent, commitment 55/55 percent, vigor 89/85 percent, dedication 94/94 percent, absorption 78/78 percent, work–home facilitation 39/34 percent, and work–home conflict 17/15 percent. ** *p* < 0.01 *** *p* < 0.04.

## Data Availability

The ARK data are free and available for all researchers who wants to do research on employees in the Academia. Information on the variables is found here: https://hunt-db.medisin.ntnu.no/ark/#instru164 or contact author for more information.

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
