# Peer review of "Antecedents and Consequences of Perceived Inclusion in Academia"

_ijerph, 2021, doi:10.3390/ijerph19010431_

Round 1
Reviewer 1 Report
The article analyzes a topic of great interest: the perception of inclusion in Norwegian universities. The theme is also reasonably relevant to the journal.
The methodology used is adequate for the proposed objectives.
Here are some questions that I hope may be of use to the authors:
1. Universities and university colleges
The abstract indicates that the participants belong to "universities and university colleges".
The journal is focused on an international scope. Some potential readers may not be familiar with the differences between these two types of institutions.
It is recommended that, if the authors consider it relevant, the difference between these two types of institutions be explained.
Furthermore, in case the authors think that there is a theoretical reason to suspect differences in the perception of inclusion between these types of institutions, it is recommended that they analyze the data to find out.
2. Gender differences in perception of inclusion.
From my point of view, the most striking result is the differences between men and women in the perception of inclusion.
I think this result needs to be discussed in more detail. The authors summarize the idea in a very powerful phrase: "unawareness of the barriers associated with being a member of a minority group might create a perception of more inclusion and fairness among men".
I think it is quite possible that this idea is true. Therefore, I think it should be explained in more detail. Likewise, this idea should be reflected in the implications of the results of the article. It should be recommended to delve into the analysis of this result and look for tools to identify and reduce the factors that are leading to this different perception between men and women.
3. Actuality of references.
In the article there are 57 references. Of these, only 9 (15.79%) have been published in the last 5 years.
It is recommended to do a search for recent references that can theoretically enrich the article and/or that are useful to compare and discuss the results obtained.
The article is correct from a theoretical point of view. But it is possible that recently works have been published that can enrich it.
4. Formal issues:
4.1. Keywords must be in alphabetic order.
4.2. The format of the references must adapt to the standards of the journal.
4.3. Lines 74-76.
"The proposed model, integrating determinants and consequences of inclusion, is outlined in Figure 1."
Figure 1 appears on page 12.
I suggest to briefly explain the model on lines 74-76. This saves the reader from having to wait until page 12 to learn about the elements of the model.
4.4. Page 19. Lines 15-16.
"aspects of leadership behavior (person r ranging from .75 to .85, see Table 1)".
Erratum. Replace "person" by "Pearson".
5. Figure 1.
The size of the figure must be adapted to the proper format of a scientific journal.
Author Response
Thank you for your constructive comments. We have changed the manuscript accordingly and marked the changes in yellow in the manuscript. Please see a point-by-point response to your comment attachment

Reviewer 2 Report
In this paper the authors conduct structural equation modelling analysis to investigate some antecedents and consequences related to perceived inclusion for jobs in academia. They authors use a large scale employee survey in Norway to test their hypotheses.
Overall I think the analysis has been conducted throroughly and appropriately on a topic that can be of interest for certain audiences.
What I would be interested to learn more about is what you can say about the direction of causality. In Subsection 1.1 when developing your hypotheses, you talk about factors being “positively related”, but avoid talking about directionality of effects. My understanding is that you do this on purpose because you cannot say with sufficient certainty, which directionality the effects have. Obviously, this significantly limits what we can learn from finding these correlations. Maybe a more inclusive environment fosters organisational commitment, maybe an organisation that has more committed employees is one that is also more inclusive. I wonder what the policy implication would be from found correlations without knowing the directionality. Also, can we comfortably represent the relationship as presented in Figure 1, which proposes a relationship at which the leader (and curiously only the leader in the given model) influences inclusiveness and through that shapes commitment, vigour, etc?
Relatedly, it may be insightful to also investigate alternative functional forms. The paper assumes that inclusiveness mediates leadership effects. However, maybe an empowering leadership has a direct effect on employees’ dedication, without the mediation of inclusiveness. Have you investigated these or other alternative functional forms?
As it currently stands, I find the introduction, and especially the literature discussion too unfocused and overburdening. As a reader I get a tremendous amount of theories and concepts thrown at, followed by a hypothesis. I would propose to have a critical look at the hypotheses section, what elements are really relevant for the paper, and what can be left out.
Specific comments, ordered as I come across in the paper, not in importance.
- Page 6 hypotheses 4 and 5. To me the definition of work-home conflict and work-home facilitation have not become entirely clear yet. In specific, it appears as if the two were just the opposite of each other. If so, I would obviously advise to only keep one of the two. If they are different enough concepts, please make this more clear.
- Page 6, H7: If men perceive their organisation on average as more inclusive than women, this should only create a parallel shift in the data. In order to find stronger correlations, men would also need to care more about the factors described in H1-H5. Otherwise, you just find a parallel shift in inclusivity perception, as hypothesised in H6.
- Page 7, lines 315-320: Instead of discussing the internal metrics of the sample (i.e. gender and age composition), I think the more interesting question may be, how the sample compares to the population in terms of age, gender and job function type.
- Page 17, line 4-5: If there are differences between the male and female parameters in the constrained model from Table 4, what are they driven by, given that parameters should be equal? Is this due to sample size then?
- Page 20, lines 89-91. I wonder if the results are robust when only considering scientific staff and PhD's. Also, iIncluding leaders themselves in the data may be a bit odd, as they basically rate their own performance. Also, in academia, leaders are often scientific staff themselves. So I would suppose the leader category pertains to staff in a uniquely management oriented position only.
- Page 22, 5 Conclusions: This could be part of the discussion or left out. This section does not really add to the manuscript.
Editorial comments, ordered as I come across in the paper, not in importance.
- Abstract: On line 9 “of rather new concept” I think there is a missing word, typo, please proofread. Also, The distinction between leader and supervisor in the abstract is not intuitively clear to me. Relatedly, throughout the article, I think you use the terms supervisor and leader interchangeably. Do you treat these as synonymous? It may be worthwhile sticking to one version throughout.
- Page 2 line 60 “Shore and colleagues: should probably simply be Shore et al. This happened also at a few other places in the manuscript. Please double check.
- Page 3 line 129 The use of past tense in “suggested” conveys that the findings may in fact not suggest these similarities anymore. That is not what you want to say, though, I think. I think present tense is more appropriate here.
- Page 4 line 193 “thrust” should be “trust”, typo.
- Page 5 line 246: Something is off with the sentence construction here. I think “women consider” should be “women who consider”, or “women considering to leave”. Please proofread.
- Page 6, lines 276-279: Please proofread sentence.
- Page 6, line 306: Please avoid headlines at the bottom of a page.
- Page 7, lines 309-310: Is this Innstrand and Christensen (2020) or is it missing from the references list? Also, for me, the doi of Innstrand and Christensen (2020) did not work, by the way. Please double check.
- Page 7, line 336: after 1, a blank space is missing.
- Page 15-16, Table 3: Please avoid breaking the table across pages.
- Page 17, line 19: “about 17 and 17 percent” Please proofread. I think R2 is not in Table 4, is it?
- Page 19, line 15-16 “person” should be “Pearson”, typo
Author Response

(The authors gave the same response as above.)
